# Increased thermal suitability elevates the risk of dengue transmission across the mid hills of Nepal

**Bipin Kumar Acharya**[1,2,3]*, **Laxman Khanal**[4], **Meghnath Dhimal**[5]

**1** Planetary Health Research Center, Kathmandu, Nepal, **2** Nepal Open University, Lalitpur, Nepal, **3** Nepal Geographical Society, Kathmandu, Nepal, **4** Central Department of Zoology, Institute of Science and Technology, Tribhuvan University, Kathmandu, Nepal, **5** Nepal Health Research Council, Kathmandu, Nepal

\* acharyageog@gmail.com

## Abstract

The burden of climate-sensitive, mosquito-borne diseases, including dengue, has significantly increased in recent years. Understanding the temporal and spatial variations of these diseases is essential for effectively controlling potential outbreaks. In this study, we utilized Moderate Resolution Imaging Spectroradiometer (MODIS) satellite land surface temperature (LST) data (MOD11A2) and a temperature-dependent mechanistic model ($R_0$) to predict the monthly suitability for dengue transmission in Nepal from 2001 to 2020 for both mosquito vectors, *Aedes aegypti* and *Ae. albopictus*. We divided the study period into two episodes: 2001–2010, which we characterized as the period of dengue emergence, and 2011–2020, identified as the period of rapid expansion. We compared the thermal suitability across these two time periods. The results indicated that approximately half of Nepal is thermally suitable for dengue transmission for at least one month, with the maximum transmission risk lasting up to nine months each year, a trend that has more or less remained stable over the past 20 years. However, strong temporal dynamics were observed in the hilly regions and around major urban centers such as Kathmandu and Pokhara, where the length of thermal suitability extended up to six months for both vector species. Consequently, the population exposed to thermal suitability increased significantly on a monthly basis. Compared to the emergence period, the proportion of the population exposed to a suitable thermal environment for six months or longer each year increased by 18% for *Ae. aegypti* and 20% for *Ae. albopictus*. These findings provide evidence-based insights that could assist health authorities in the control and management of dengue in Nepal.

## 1. Introduction

Dengue (break-bone fever) is a vector-borne arboviral infection primarily transmitted by the *Aedes aegypti* and *Ae. albopictus* mosquitos [1]. It has been a leading public health challenge worldwide causing approximately 390 million infections annually and becoming endemic in over 100 countries [2]. The incidence of dengue has surged dramatically over the past three

**Data availability statement:** The geospatial dataset including MODIS LST (https://lpdaac.

usgs.gov/), grided population data (https://hub.worldpop.org/geodata/summary?id=27800), shapefile of district boundary of Nepal are open access and can be freely downloaded (https://nationalgeoportal.gov.np/) from the provided geoportal. The geolocation data of dengue occurrence location is available in S1 Table.

**Funding:** The author(s) received no specific funding for this work.

**Competing interests:** The authors have declared that no competing interests exist.

**Abbreviation:** MODIS, moderate resolution imaging spectroradiometer; LST, land surface temperature; AUC, area under the curve

decades, accompanied by a rapid geographic expansion [3,4]. Given the effects of global climate change, increased human mobility, and rapid urbanization, further spread and rising incidence are anticipated in the future [5–7]. Currently, there are no effective vaccines or specific therapies available to curb the rapid global spread of dengue [8]. Therefore, understanding the dynamics and spatial distribution patterns of dengue is crucial for developing effective control interventions.

Dengue transmission is highly sensitive to temperature which is a key environmental factor influencing the disease dynamics and distribution [9,10]. Temperature variations affect not only the survival and development of mosquito vectors but also biting rates and vector-host interactions which influence propagation of the virus. Higher temperatures can enhance vectorial capacity by shortening mosquito development time, increasing vector competence, and reducing the extrinsic incubation period [11,12]. However, thermal extremes can negatively impact disease dynamics [6]. For instance, a temperature range of 14–15°C is critical for the survival of adult *Ae. aegypti* mosquitoes; temperatures below this threshold limit their mobility and ability to feed on blood [13]. Extreme temperatures, such as those below 16 °C or above 36 °C, can significantly decrease adult mosquito longevity and female fecundity [14]. Very low temperature limits not only egg hatching and larval development [15] but also extrinsic incubation period and viral development rate [16]. Winter temperature is a key limiting factor for the persistence of *Ae. aegypti* eggs in the environment [17] because freezing temperatures can destroy mosquito eggs and larvae [18]. A land surface temperature (LST) as low as 13.8 °C in winter has been identified as critical for *Ae. aegypti* larvae, potentially leading to their near disappearance in subtropical regions of Taiwan during the East Asian winter monsoon [19]. Because of being a strong driver, temperature variable has been widely used in mapping the mosquito borne diseases such as dengue, malaria and their mosquito vectors [20–22]. Previous studies have successfully used MODIS LST in mapping the potential distribution of *Aedes* mosquito [19,23,24] across the world based on the temperature constraints [25].

While the incidence and distribution of vector-borne diseases are often influenced by the abundance and distribution of their primary vectors, challenges remain in understanding disease dynamics. A major issue in spatial epidemiological risk modeling is the availability of high-resolution spatial temperature data, particularly in complex terrains with sparsely distributed meteorological stations [23]. Remote sensing satellite-based LST data can address this gap. Advances in remote sensing technology provide opportunities to collect environmental data, enabling the quantification of spatial variations in disease distribution at fine spatial, temporal, and spectral resolutions [26]. These satellite systems improve data availability for scientific purposes and predictive epidemiological studies [27]. Remote sensing can measure spatial variation in temperature more accurately in regions with wide elevation variations where interpolation may introduce error [28]. The key instrument on board both the Terra and Aqua satellites is the Moderate Resolution Imaging Spectroradiometer (MODIS). Twice a day each MODIS sensor delivers global coverage at 250 m, 500 m and 1000 m resolutions in different spectral bands in a hierarchical spatial scheme [29]. The MODIS instruments capture data on 36 spectral bands that range in both wavelength and spatial resolution and have been used in study of disease vector population dynamics. Increased computing power and spatial modeling capabilities of geographic information systems could extend the use of remote sensing beyond the research community into operational disease surveillance and control [26].

Dengue has rapidly expanded in Nepal, now affecting all 77 districts, with significant caseloads and disease burden. The first reported case of dengue in 2006 was from Chitwan, a lowland district in central Nepal [30]. Initially, dengue was sporadic, confined mainly to a few districts in the Tarai region. However, since 2010, dengue cases have been rising continuously,

leading to large outbreaks in 2010, 2013, 2016, and 2019, and even reaching higher-altitude mountain districts, confirming the expanding geographical range of the disease [31–33]. In 2019, Nepal experienced a notable outbreak with 17,992 reported cases and six deaths, representing over 72% of cumulative cases in the country. While *Ae. albopictus* has been endemic in Nepal since the 1950s, *Ae. aegypti* has recently spread throughout the country and has been linked to the recent dengue outbreaks [34–36]. All four serotypes of dengue virus are circulating in Nepal since 2006. Despite this, there is limited information on the spatial risk patterns of dengue in Nepal, largely due to a lack of spatially explicit research. Although some attempts have been made to map dengue incidence using district-level aggregated data, rigorous spatiotemporal predictions using the Earth observation data remain scarce [37,38]. Acharya et al. [39] assessed climate factors associated with dengue in Nepal, but it relied on interpolated climate surfaces based on data from the 1960s to 1990s, predating the emergence of dengue in the country. Additionally, there is insufficient knowledge about the length of the dengue transmission season in Nepal, which is crucial for timely surveillance and control efforts.

This study aimed to assess the thermal suitability for dengue in Nepal and analyze its changing patterns over the last 20 years using satellite-based LST data for both vectors, *Ae. aegypti* and *Ae. albopictus*. By integrating temperature-dependent mechanistic models, we utilized high temporal resolution MODIS LST time series from 2001 to 2020 to represent the emergence (2001–2010) and expansion (2011–2020) of dengue in Nepal. The resulting predictive maps provide valuable resources for health authorities in managing and controlling dengue transmission. Furthermore, this study has estimated the population at risk of dengue based on different lengths of the transmission season.

## 2. Materials and methods

### 2.1. Study area

Nepal lies in southern side of central Himalayas between China and India at 26° to 30° latitudes north and 80° to 88° longitudes east (Fig 1). There are remarkable variations in land topography where elevation ranges from 60 m to 8848 m within shorter than 200 km north-south extent. Based on the elevation and land topography, Nepal is divided into five major physiographic zones- Tarai (below 600 m), Siwalik (100–2000 m), Hill (200–3500 m), Middle Mountain (700–4100 m) and High Mountain (1800–8800 m) [40]. In a broader sense, the climate of Nepal is subtropical monsoon climate with distinct seasonality in temperature, precipitation and humidity. Four distinct seasons in Nepal include winter (December of previous year, January, and February), pre-monsoon (March–May), monsoon (June–September), and post monsoon (October and November) [41]. The extreme variation in altitude and topography has also resulted in to four bioclimatic zones, namely warm temperate, temperate, cool temperate and cold climate [41]. The topographic orientation and climatic variation have great impacts on population distribution, infrastructure development and on the distribution of pathogen and vectors in Nepal.

### 2.2. Data and processing

The overall methodology of this study is depicted in Fig 2 which is followed by a detailed description.

We downloaded MOD11A2 version 6.0 spanning from 2001 to 2020 from the Land Processes Distributed Active Archive Center (LPDAAC/NASA, https://lpdaac.usgs.gov) using the MODIStsp package [42] in R statistical software [43]. Within the MODIStsp framework, we extracted Tagged Image File (TIF) layers from originally supplied Hierarchical Data Format (HDF) file, mosaicked two MODIS tile (h25v5 and h2v6), re-projected from sinusoidal

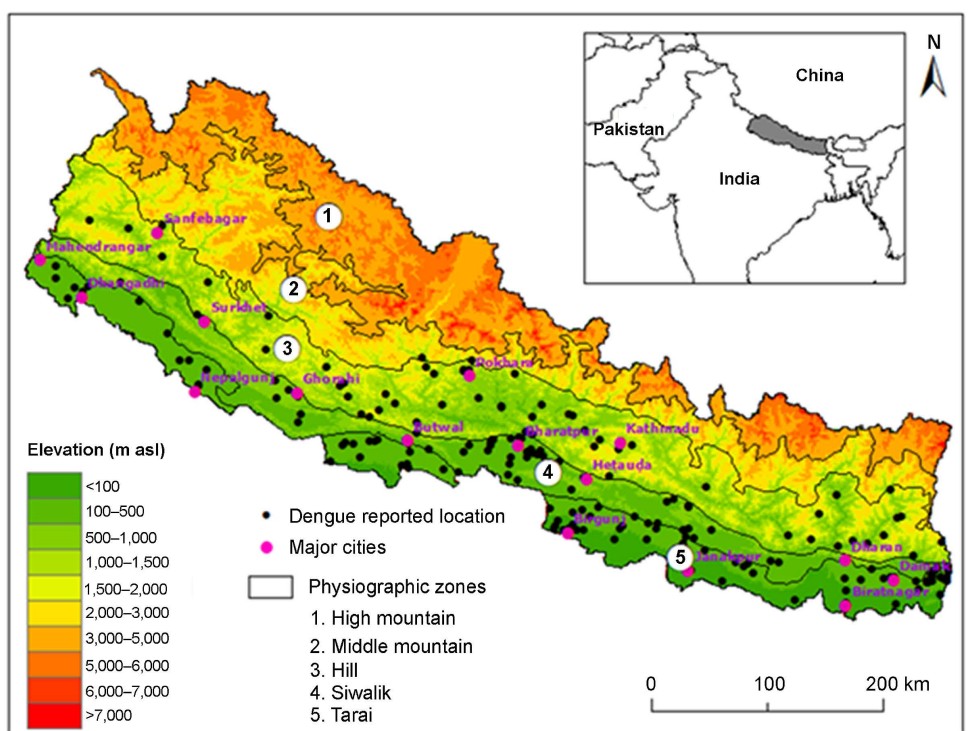

**Fig 1. Map of Nepal showing elevational gradient, physiographic zones and dengue reported locations.**

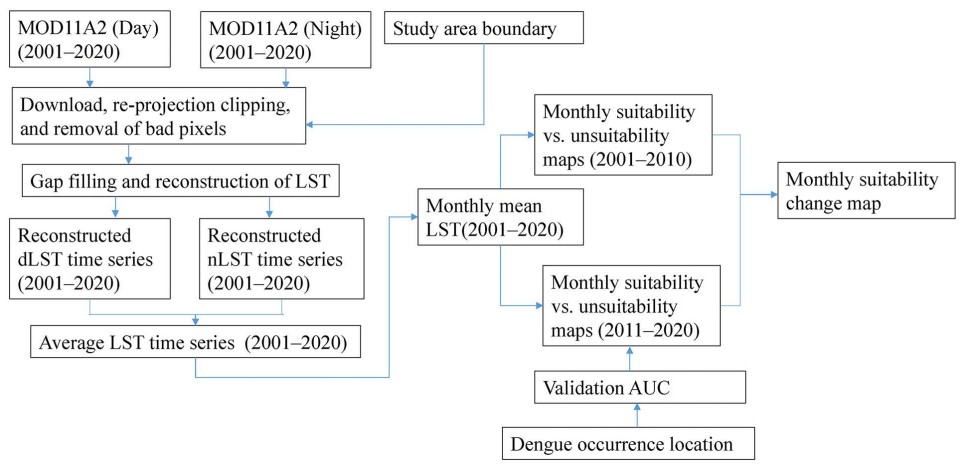

**Fig 2. Methodological workflow of the study.**

projection system to Universal Transverse Mercator (UTM zone 45 north) and removed contaminated pixels using the quality assurance layers supplied with MODIS product then clipped each layer by the country boundary of Nepal. Finally, we converted Kelvin to degree centigrade by subtracting 273.15 from each pixel value of the time series. We processed altogether 72822 MODIS tiles for 20 years period from 2001/01/01–2020/12/30 representing two decade of dengue emergence and expansion in Nepal.

The preprocessed MODIS LST time series cannot be directly used as an input in mapping and modelling because of data loss due to cloud contamination, orbit convergence resulting in the loss of data. Altogether we observed 294680718 void pixels in our space and time domain accounting for around 10 percent of total data loss. If such incomplete data is used, the results would be skewed in either direction [23]. To overcome this problem, the Data Interpolating Empirical Orthogonal Functions (DINEOF) interpolation method was adopted to reconstruct the LST. The DINEOF method proposed by Beckers and Rixen [44] has been successfully used in a number of previous studies including sea surface temperature reconstruction [45–47], land surface temperature reconstruction [48] and reconstruction of fraction of green vegetation [49]. The DINEOF is developed based on the Empirical Orthogonal Functions to solve the missing data problem appearing in many geoscience fields. Compared to traditional interpolation methods, the DINEOF requires fewer input parameters, has high computational efficiency and does not necessitate the calculation of spatial or temporal correlation. The DINEOF also has no strict assumption on the distribution of the original dataset. Therefore, it could be applied even under high missing data rate condition [47]. We used 'sinker' package in R statistical software to implement DINEOF function and reconstruct the LST time series. The reconstructed LST was evaluated using Root Mean Square Error (RMSE) metric where the achieved overall accuracy was 1.6. Finally, day and nighttime LST time series were combined to obtain the average LST which was later used to compute two decades average LST dividing the entire study period into two phases, 2001–2010 (the period of dengue emergence in Nepal) and 2011–2020 (the period of rapid expansion in Nepal).

## 2.3. Modelling and mapping

We used temperature dependent $R_0$ model to derive optimum temperature range and map the thermal suitability in Nepal. The mapping approach is the use of basic cutoff for the thermal interval where viral transmission is possible. Similar approach to map the distribution of dengue and its mosquito vector in regional [22,50] and global [21] scales have been proven well.

This mechanistic model uses temperature-based traits of vector mosquito, virus and host population to estimate the basic reproduction number ($R_0$). $R_0$ refers to the number of secondary infections that originate from single infected individual to the fully susceptible population [10]. $R_0$ values greater than 1 correspond to epidemic growth, and an $R_0$ of 1 corresponds to endemicity. $R_0$ values less than 1 correspond to transmission ceasing as the number of new infections decreases in subsequent generations. The model is expressed as:

$$R_0\left(T\right) = \left( \frac{a^2 \star b \star C \star e \dfrac{\mu}{PDR} \star EFD \star P_{EA} \star MDR}{N \star R \star \mu^3} \right)^{1/2}$$

In this equation, (T) indicates trait which is a function of temperature; $a$ is the per-mosquito biting rate, $b*c$ is vector competence, $\mu$ is the adult mosquito mortality, $EFD$ is the number of eggs produced, $P_{EA}$ is probability of mosquito survival from egg to adult, $MDR$ is the mosquito immature development rate, $N$ is human population size; and $r$ is human recovery rate per female mosquito per day. However, temperatures at a larger spatial scale vary extensively and influence the calculation and interpretation of $R_0$ [51,52]. Therefore, in the vectorial model, $R_0$ is viewed as a simple metric for assessing the relative suitability of temperature for transmission, rather than as an absolute measure for secondary case distributions, invasion and extinction thresholds, or expected equilibrium prevalence [53,54]. Mordecai et al. [10] parameterized the $R_0$ model in a Bayesian framework to account for uncertainty in the experiential

data and validated it based on human case data for three viruses (dengue, chikungunya, and Zika). The advantage of Relative $R_0$ approach is that it allows to estimate the thermal optimum and limits, at which $R_0$ is maximized or goes to zero, respectively [10]. Outside this thermally suitable range, transmission becomes impossible because one or more processes essential for transmission have completely ceased [55].

Following the Mordecai et al. [10] we chose optimal temperature range parameterized at 97 percent confidential interval on Bayesian framework (number of months for which $R_0>1$) to classify 240 monthly average LST raster layers of 20 years into suitability versus unsuitability based on the separate optimal temperature range (*Ae. aegypti*, 21.3–34.0 °C, and for *Ae. albopictus*, 19.9–29.4 °C). The 240 monthly binary maps were then aggregated by simple arithmetic operation and to obtain number of months suitable for disease transmission for the two temporal windows. The resulting maps were visualized using Arc GIS v.10.5.

To estimate the exposed human population in thermal suitability and assess the burden of dengue in Nepal, we retrieved 100-meter spatial resolution grided population data of 2010 for the period of emergence and 2020 for the period of expansion from the WorldPop geoportal (https://www.worldpop.org/geodata/listing?id=78) which was later aggregated in 1000 m pixel to match spatial resolution of both raster layers. Then, we overlaid both raster layers and extracted the population of each pixel. To assess altitudinal shift in spatial distribution of thermal suitability and assess the monthly exposure of human population, we downloaded SRTM DEM in 90 m spatial resolution (https://srtm.csi.cgiar.org/) and overlaid with dengue risk map.

## 2.4. Model validation

The output maps of both vector species for the period of expansion were validated against the reported location of dengue using the Area Under Curve (AUC) of Receiver Operating Characteristic (ROC). The AUC measures the predictive performance of the model by comparing the model's predictive ability to the random prediction [39]. The AUC value ranges from 0 to 1 where 0.5 indicates random prediction and higher values correspond to a better model prediction. The reported dengue occurrence locations were collected from two different sources. The first source was earlier compilation of 124 occurrence locations, which represent report of dengue cases at least one time between 2010–2014 from these locations [39]. The reported dengue occurrence locations for the later period 2010–2020 were collected from Epidemiology and Disease Control Division (EDCD) of the Government of Nepal at local address level known as a 'line listing file'. The address level data were then geocoded using 'opencage' geocoder package (https://opencagedata.com/) in R. Finally 149 occurence location were prepared for the validation purpose (S1 Table). In addition to occurrence location, absence location; representing the locations less likely of dengue transmission to human were generated. As we did not have the absence location data, we used pseudo-absence, not the real absence records [56]. We generated 300 pseudo-absence data covering the entire country using the randomPoints () function of 'dismo' package in R. We made it sure that our background points were sufficiently away from the presence points [57]. The AUC was computed using pROC package in R.

## 3. Results

## 3.1. Model validation

The model validation results show that the optimum thermal range (*Ae. aegypti*, 21.3–34.0 °C, and for *Ae. albopictus*, 19.9–29.4 °C) implemented in the prediction of number of months suitable for dengue transmission is moderately high with AUC value 0.8 for *Ae. aegypti* and 0.77 for *Ae. albopictus* (Fig 3). The predicted accuracy is slightly higher for *Ae. aegypti* compared to *Ae. albopictu*s, however, both the AUC values indicate the robustness of the models.

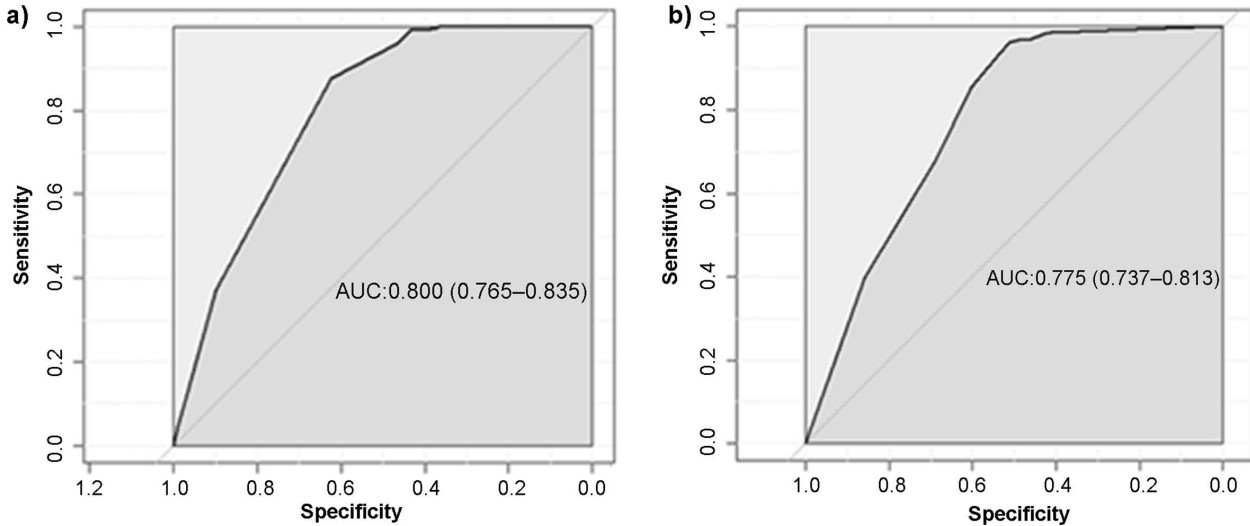

**Fig 3. Area under curve (AUC) of a)** *Aedes aegypti* **and b)** *Ae. albopictus* **during the period of emergence.** Values inside the brackets indicate the confidence intervals.

## 3.2. Spatial distribution patterns of thermal suitability of the dengue vectors

The spatial distribution of thermally suitable areas for *Ae. albopictus* and *Ae. aegypti* in Nepal is presented in Fig 4. The thermally suitable areas of dengue vectors were extensively

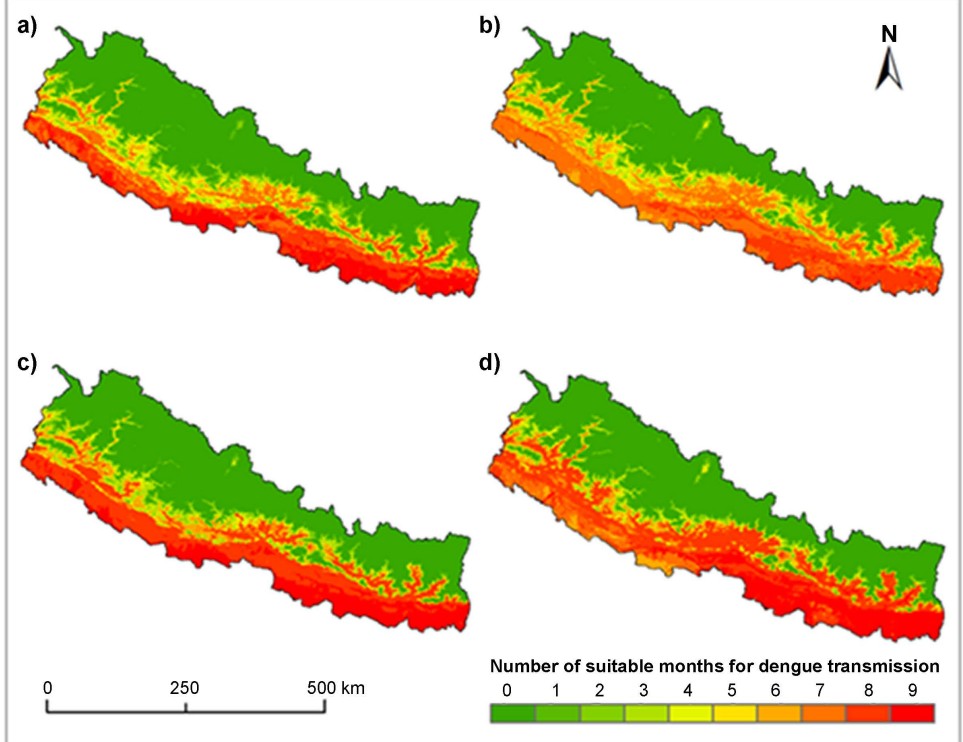

**Fig 4. Spatial distribution of predicted thermal suitability based on temperature threshold for a)** *Aedes aegypti* **and b)** *Ae. albopictus* **during 2001–2010; and c)** *Ae. aegypti* **and d)** *Ae. albopictus* **during 2011–2020.**

distributed in southern low land including Tarai and Siwalik region across Nepal. In addition, the thermal suitability *for the* transmission of both species has been extended in low elevated river valleys of mid hills. Such extension is more apparent in central Nepal than other areas.

Geographic projection of $R_0$ (T) model predicted about 53% areas of Nepal as thermally suitable for dengue transmission by *Ae. aegypti* and 57% for *Ae. albopictus* for at least a month of a year in 2011–2020 which was slightly lower (52% and 56%) during the 2001–2010 (Fig 5). The proportion of suitable areas for the transmission for 6 months or longer was 29% for *Ae. aegypti* and 32% *for Ae. albopictus* during 2011–2020. For 2001–2010, it was 24% and 19% for *Ae. aegypti* and *Ae. albopictus*, respectively.

We observed a significant change in the length of thermally suitable months of dengue transmission during the last 20 years in Nepal. The temporal transmission window of *Ae. aegypti* extended in 24.51% area of the country during the period of expansion compared to the period of emergence. The number of extended months ranged from one to seven but

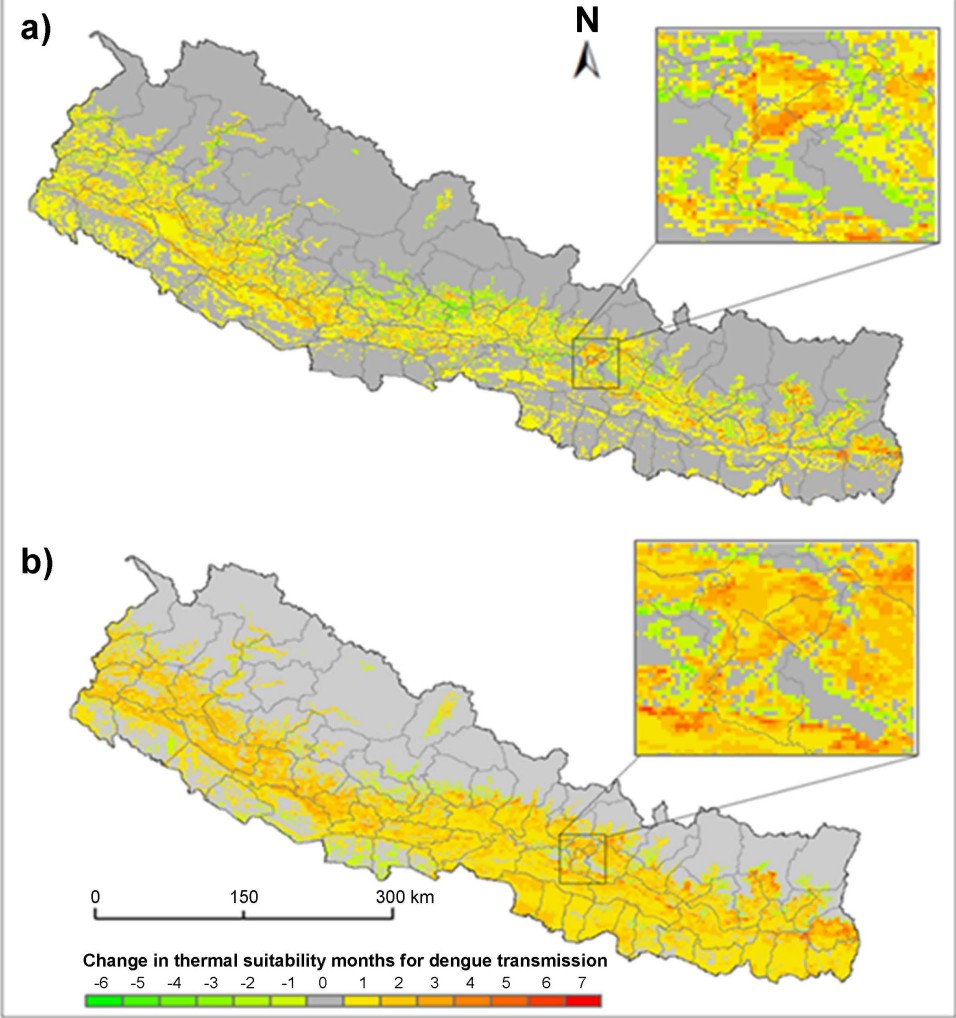

**Fig 5. Change in thermal suitability between 2001–2010 and 2011–2020 for a)** *Ae. aegypti* **and for b)** *Ae. albopictus*. The highlighted inset shows the thermal suitability in and around the Kathmandu Valley.

in the majority areas the extension was either one or two months. For the *Ae. albopictus* the temporal extension was observed in 33.38% area of the country.

The length of transmission season of dengue not only extended but also decreased in some areas of the country. The proportion of areas where the transmission was decreased was about 5% of the total area for both mosquito species. The notable temporal changes on the thermal suitability were along the edge of suitable areas across the mid hills of Nepal. The clear spatial change was also observed around the mega urban centers. For example, thermal suitability around the Kathmandu Valley has been increased up to 5 months especially in the recently urbanized surroundings of the core center.

## 3.3. Elevation gradient and transmission risk

Elevation is an important gradient in the distribution of thermally suitable areas of dengue transmission in Nepal. The southern lowland Tarai is mainly endemic to *Aedes* mosquitos and is suitable for dengue transmission which gradually decreases with increasing elevation from south to north. Our results showed that about 42% of thermally suitable areas lie below 500 m asl for both mosquito species which gradually decline with increasing elevation from south to north (Table 1).

The spatial distribution of thermally suitable areas has decreased in elevation below 500 m but has increased in the elevation range between 500 m to 1500 m asl for vector both vector species. However, in other elevation zones it is relatively constant in both time periods and for both mosquito species.

## 3.4. Population at risk

Population exposure analysis reveals that about 93% of population lives in areas thermally suitable for dengue transmission at least a month of year (Table 2) by *Ae. aegypti* mosquito while this proportion reaches around 98% for the *Ae. albopictus*. This proportion was lower in 2010; 88.76% for *Ae. aegypti mosquito* and 93.43% *Ae. albopictus*.

The population exposure by months has increased significantly in the last 20 years. For example, the proportion of population exposed in thermally suitable environment for 6 months or longer was 72% for *Ae. aegypti* and 74% for *Ae. albopictus* in 2010 but reached 90% and 94% respectively in 2020. These figures go higher and with apparent change when taken with 3 months transmission areas where 83% for *Ae. aegypti* and 87% for *Ae. albopictus* increased to 87% and 96%, respectively.

**Table 1. Distribution of temperature suitability areas of for dengue transmission along the elevation gradient in Nepal.**

| Elevation | 2001-2010 | | | | 2011-2020 | | | |
|---|---|---|---|---|---|---|---|---|
| | *Ae. aegypti* | | *Ae. albopictus* | | *Ae. aegypti* | | *Ae. albopictus* | |
| | Area (%) | Cumulative Area (%) | Area (%) | Cumulative Area (%) | Area (%) | Cumulative Area (%) | Area (%) | Cumulative Area (%) |
| 0–500 | 42.01 | 42.01 | 39.15 | 39.15 | 42.90 | 42.90 | 39.72 | 39.72 |
| 500–1000 | 23.92 | 65.93 | 22.38 | 61.53 | 24.47 | 67.37 | 22.72 | 62.44 |
| 1000–1500 | 21.96 | 87.89 | 21.70 | 83.22 | 21.76 | 89.13 | 21.63 | 84.08 |
| 1500–2000 | 10.28 | 98.17 | 12.70 | 95.93 | 9.46 | 98.59 | 12.49 | 96.56 |
| 2000–2500 | 1.32 | 99.49 | 2.97 | 98.90 | 1.06 | 99.64 | 2.60 | 99.17 |
| 2500–3000 | 0.10 | 99.59 | 0.35 | 99.25 | 0.06 | 99.71 | 0.27 | 99.44 |
| 3000–3500 | 0.16 | 99.75 | 0.22 | 99.47 | 0.11 | 99.81 | 0.16 | 99.60 |
| 3500–4000 | 0.22 | 100.00 | 0.41 | 100.00 | 0.18 | 100.00 | 0.40 | 100.00 |

**Table 2. Proportion of population living in different months of thermal suitability of dengue transmission for the year 2010 and 2020 in Nepal.**

| Suitability duration (months) | *Ae. aegypti* | | | | *Ae. albopictus* | | | |
|---|---|---|---|---|---|---|---|---|
| | 2010 | | 2020 | | 2010 | | 2020 | |
| | Population (%) | Cumulative (%) | Population (%) | Cumulative (%) | Population (%) | Cumulative (%) | Population (%) | Cumulative (%) |
| 9 | 39.77 | 39.77 | 53.54 | 53.54 | 0.85 | 0.85 | 38.78 | 38.78 |
| 8 | 18.61 | 58.38 | 19.01 | 72.55 | 25.25 | 26.1 | 28.04 | 66.83 |
| 7 | 5.74 | 64.12 | 8.88 | 81.43 | 29.73 | 55.83 | 18.46 | 85.29 |
| 6 | 8.54 | 72.65 | 8.62 | 90.05 | 18.52 | 74.35 | 9.44 | 94.73 |
| 5 | 1.83 | 74.48 | 1.22 | 91.27 | 4.8 | 79.14 | 0.56 | 95.29 |
| 4 | 2.26 | 76.74 | 1.14 | 92.41 | 3.92 | 83.06 | 0.68 | 95.97 |
| 3 | 6.42 | 83.15 | 1.67 | 94.08 | 4.8 | 87.85 | 1.02 | 96.99 |
| 2 | 3.80 | 86.95 | 2.27 | 96.35 | 2.57 | 90.42 | 0.75 | 97.73 |
| 1 | 1.80 | 88.76 | 0.47 | 96.82 | 3.01 | 93.43 | 0.31 | 98.04 |

## 4. Discussion

This study highlights the significance of earth observation satellite data in mapping the thermal suitability of two mosquito vector species of dengue, *Aedes aegypti* and *Ae. albopictus*, in Nepal. We integrated the optimal temperature range, derived from a temperature-dependent mechanistic model $R_0(T)$, with gap-filled MODIS Land Surface Temperature (LST) time series data from 2001 to 2020. The study period was divided into two phases: 2001–2010 as the period of emergence and 2011–2020 as the period of rapid expansion. Prediction assessments based on the Receiver Operating Characteristic (ROC) curve indicated that the selected optimal temperature range accurately predicted suitability, achieving an Area Under the Curve (AUC) value above 0.75. The results demonstrated that temperature is the primary driver of spatial variation in dengue risk across Nepal. The months of thermal suitability for both mosquito vectors have notably expanded, particularly in elevation ranges between 500 and 1500 meters above mean sea level. As a result, the proportion of the population exposed to thermal suitability per month has significantly increased. Our findings could be instrumental in revising and updating dengue surveillance and control strategies, particularly targeting the mid-hills and recently expanded urban areas.

We found that year-round transmission of dengue does not occur in Nepal due to seasonal variation of weather. Even in the less elevated Tarai and Siwalik regions which are considered as endemic to both mosquito vectors [58], the maximum length of transmission goes only up to 9 months. The outbreak usually starts following the monsoon (June–September) and peaks in the post monsoon (October) season and declines with the onset of winter [37]. Winter season is usually free from dengue due to the fall of temperature. Seasonal variation in the length of transmission is concurrent with previous entomological study [58]. This study, for the first time quantified cumulative months' thermal suitability of dengue transmission in Nepal. The length of transmission is normally longer in lowlands and shorter in highlands of Nepal. Due to extreme cold, northern high mountain areas are unsuitable for dengue transmission throughout the year.

Elevation is a key determinant of the spatial distribution of temperature-sensitive arboviral diseases [59]. We also observed strong effects of elevation in the spatial distribution of length of transmission of dengue in Nepal. As a result, the proportion of area and length of transmission for both mosquito vectors decrease with increasing elevation. However, we observed clear expansion of thermally suitable areas between the elevation 500–1500 m asl but shrinkage below 500 meters between the period of emergence and the period of expansion. Therefore,

mid hills region lying between 500 and 1500 m asl elevation are critical for potential dengue outbreaks. If outbreak occurs in these areas, situation could be devastating due to immuno-logically naive population, poor public health infrastructure and lack of previous experience in managing the similar disease outbreaks [60]. Therefore, regular surveillance of both dengue cases and mosquito vectors is essential in these areas.

Our study quantified extensive areas that are thermally suitable for dengue transmission covering the entire Tarai and Siwalik regions. More than 50 percent of Nepal is thermally suitable for dengue transmission at least a month of a year which is significantly higher compared to the previous study [39]. The proportion of thermally suitable areas for 6 months or longer is 33% which is much higher compared to the previous study. The discrepancies could be due to methodological differences including prediction method and variable taken into consideration. For instance, previous study was based on statistical methods and temperature and precipitation variables. But the current study followed a mechanistic approach based on thermal traits of mosquito vectors, the virus, and susceptible human population. However, it is also equally likely that under prediction could be due to missing of occurrence record because of poor surveillance in these regions. Ryan et al. [21] also predicted several places of temperate zone suitable for dengue outbreak up to six months that were previously considered as unsuitable areas.

Despite continuous reports of new cases and recent massive outbreak [61], most populated urban centers such as Kathmandu, Pokhara had not been mapped as climatically suitable areas of dengue transmission. However, our results identified both cites as thermally suitable for 6 months or longer of a year illustrating the importance of process based mechanistic spatial prediction of disease risk especially in the data scarce region like Nepal. These high population density areas with poor sanitations augmented by the rise in temperature suffer from a higher risk of dengue transmission.

Spatial patterns of thermal suitability of both vector species are broadly concurrent with each other. However, little variation can be observed in spatial patterns due to different thermal optimum for transmission these mosquitos (29 °C vs. 26 °C). *Aedes albopictus* transmission is better suited to cooler environments than *Ae. aegypti* [10]. Therefore, *Ae. albopictus* based length of dengue transmission in western Nepal is shorter whereas the northern limit of suitability is expanded northward in the middle and high mountains.

Our study illustrates the importance of earth observation data in mapping and modeling the dengue transmission risk with sufficient accuracy. Earth observation data has several advantages over the station-based data particularly in the mountainous region like Nepal where ground station is sparsely distributed. Unlike station-based data, satellite observation provides continuous measurement over a longer period. The uncertainty and interpolation error inherited in station-based data is minimized in earth observation data [62]. However, the limitation of optical satellite like MODIS LST is data loss due to cloud contamination. In this study 294680718-pixels accounting about 10% of the pixels were void which was interpolated by DENIOF function with sufficient accuracy. This study used high resolution temporal data on dengue occurrence MODIS LST data for risk mapping purposes contributing to the field of application of remote sensing in spatial epidemiology [63].

This study has important policy implication in generating evidence-based knowledge. Evidence based knowledge on infectious disease like dengue with no vaccine or therapeutics is vital managing its impacts. Earth observations satellites provide quick and accurate datasets on the changing drivers for disease occurrence and spread which could provide a tactical advantage in predicting disease risks. The thermal suitability map developed here can support surveillance and control efforts for dengue control and management. Despite this importance, there are several methodological and empirical limitations associated to this study. The

occurrence and spread of dengue depend not only on the temperature but also on several variables including precipitation, other environmental factors, and socioeconomic level of development. But here we only illustrated the importance of temperature in spatial assessment and mapping the risk. We employed the DINEOF interpolation method to reconstruct the LST due to its ability for self-adaptation and large-area missing data reconstruction. However, this method also has limitations such as it is sensitive to the missing data and the number and distribution of the cross-validation points may affect the accuracy of the interpolation. Therefore, maps produced in this study should be interpreted as only thermal suitability. In addition, we validated predicted map based on the reported cases of dengue for both mosquito vectors together. Future studies are suggested to have validation using the occurrence record of each mosquito vector independently.

## 5. Conclusion

In this study, we assessed and mapped changing spatial and temporal patterns of thermal suitability of dengue transmission in Nepal using the gap filled MOIDS LST time series data and temperature dependent mechanistic model ($R_0$). The results revealed about half of the area of Nepal is thermally suitable for dengue transmission at least for a month with maximum transmission risk of nine months of a year. During the last 20 years (2001–2020) subtle increase in thermal suitability for dengue was observed in Nepal especially in elevation between 500–1500 while above this elevation ranges, thermal suitability remains constant and below this subtle decrease was observed. However, marked increase was observed in the temporal window across the hill and around the mega urban centers where the length of thermal suitability was found extended up to six months for both vector species indicating the future risk of outbreaks. As a result, population exposure with thermal suitability per month increased significantly. Compared with the period of emergence, the proportion of population exposed in suitable thermal environment for six months or longer increased by 18% for *Ae. aegypti* and 20% for *Ae. albopictus.* Dengue control and management within Nepal should be improved through evidence based interventions by focusing on regions at elevations between 500–1500 m and urban centers which have the most significant growth of thermal suitability and population exposure risk to dengue transmission within the past two decades.

## Supporting information

**S1 Table. Dengue occurrence location.**
(XLSX)

## Author contributions

**Conceptualization:** Bipin Kumar Acharya, Meghnath Dhimal.

**Data curation:** Bipin Kumar Acharya, Laxman Khanal.

**Formal analysis:** Bipin Kumar Acharya.

**Methodology:** Bipin Kumar Acharya.

**Software:** Bipin Kumar Acharya.

**Supervision:** Meghnath Dhimal.

**Validation:** Bipin Kumar Acharya.

**Visualization:** Bipin Kumar Acharya.

**Writing – original draft:** Bipin Kumar Acharya.

**Writing – review & editing:** Laxman Khanal, Meghnath Dhimal.

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
