## [Decision Letter · Decision Letter 0]

22 Jan 2025

PONE-D-24-56974Increased thermal suitability Elevates Risk of Dengue Transmission Across the Mid Hills of NepalPLOS ONE

Dear Dr. ACHARYA,

Thank you for submitting your manuscript to PLOS ONE. After careful consideration, we feel that it has merit but does not fully meet PLOS ONE’s publication criteria as it currently stands. Therefore, we invite you to submit a revised version of the manuscript that addresses the points raised during the review process.

Dear Authors,

This manuscript is in need of a major revision; for instance, some sentences need clarification, the reference should be cited in a consistent format, and the methods should be clear enough to allow this work to be reproducible. Please kindly address the concerns of our reviewers in order to enhance the quality and clarity of your work. Thank you!==============================

We look forward to receiving your revised manuscript.

Kind regards,

Harapan Harapan, MD, PhD

Academic Editor

PLOS ONE

2. In the online submission form, you indicated that the geospatial dataset including MODIS LST, grided population data, shapefile of Nepal are open access and can be freely downloaded . While the geolocation data of dengue occurence will be made avaiable upon the resonable request.

3.We note that Figures 1, 4, and 5 in your submission contain [map/satellite] images which may be copyrighted. All PLOS content is published under the Creative Commons Attribution License (CC BY 4.0), which means that the manuscript, images, and Supporting Information files will be freely available online, and any third party is permitted to access, download, copy, distribute, and use these materials in any way, even commercially, with proper attribution. For these reasons, we cannot publish previously copyrighted maps or satellite images created using proprietary data, such as Google software (Google Maps, Street View, and Earth). For more information, see our copyright guidelines: http://journals.plos.org/plosone/s/licenses-and-copyright.

a. You may seek permission from the original copyright holder of Figures 1, 4, and 5 to publish the content specifically under the CC BY 4.0 license.  

Reviewers' comments:

Reviewer's Responses to Questions

**Comments to the Author**

1. Is the manuscript technically sound, and do the data support the conclusions?

Reviewer #1: Yes

Reviewer #2: Yes

2. Has the statistical analysis been performed appropriately and rigorously? 

Reviewer #1: Yes

Reviewer #2: Yes

3. Have the authors made all data underlying the findings in their manuscript fully available?

Reviewer #1: Yes

Reviewer #2: Yes

4. Is the manuscript presented in an intelligible fashion and written in standard English?

Reviewer #1: Yes

Reviewer #2: Yes

5. Review Comments to the Author

Reviewer #1: This study was utilized Moderate Resolution Imaging Spectroradiometer satellite land surface temperature data (MOD11A2) and a temperature dependent mechanistic model to predict the monthly suitability and its changing pattern for dengue transmission in Nepal from 2000 to 2020 for both mosquito vectors, Aedes aegypti and Ae. albopictus. This study estimated the population at risk of dengue based on different lengths of the transmission season. The results from this study could be a model for dengue management and vectors control in Nepal and other dengue endemic countries.

However, there are some comments for this manuscript to address as follows:

1. Citations of the references should follow the author guidelines of PLOS ONE. Most of the citations presented as author name and year, but some citations presented as the number. For examples, the references on Page 9, Lines 195-196, “…………including sea surface temperature reconstruction [55–57],………, on Page 10, Lines 230 and 233………the calculation and interpretation of R0 [24–28]. and …………equilibrium prevalence [29–31].” Please check the citation of references according to the author guidelines of PLOS ONE.

2. The term “dengue fever” and “dengue” have different meaning. The meaning of “dengue fever” is the degree of disease severity whereas “dengue” stands for disease occurring by dengue viral infection. Please revise the following sentences and other sentences.

a. On Page 3, Lines 61-62, “Currently, there are no effective vaccines or specific therapies available to curb the rapid global spread of dengue fever (Simon-Lorière et al., 2017).

b. On Page 5, Line 139, “This study aimed to assess the thermal suitability for dengue fever in Nepal and analyze……….”

3. The introduction part should be shorten, concise, and has supporting information relevant to the rational of the study. For example, “A land surface temperature (LST) as low as 13.8 °C in winter has been identified as critical for Ae. aegypti larvae, potentially leading to their near disappearance in subtropical regions of Taiwan during the East Asian winter monsoon (Tsai et al., 2018) on Page 3, Lines 80-82.” and “An LST of 13.8 °C in winter has been identified as critical for the survival of Ae. aegypti larvae (Tsai et al.,2018).” On Page 4, Lines 87-89. These two sentences have similar meaning.

4. Figure 1 has four major physiographic zones, but Siwalik and Tarai have the same number of 4. However, on Page 6, Lines 152-154, “Nepal is divided into five major physiographic zones- Tarai (below 600 m), Siwalik (100–2000 m), Hill (200–3500 m), Middle Mountain (700–4100 m) and High Mountain (1800–8800 m) (LRMP, 1986).”

5. Materials and Methods :

a. Page 8, Lines 173-174, “………(R Development Core Team)”, the statistical program R should be cited correctly.

b. Page 8, Line 179, “………we extracted TIF layers from originally supplied HDF file, ….” The abbreviation for file type should be cited as full name, then followed by abbreviation.

c. Page 8, Lines 174-178, “The MOD11A2 is 8-day averaged LST product derived from MOD11A1 (Z. Wan, 2015). The MODA1 is daily LST product derived by computation from two adjacent thermal infrared bands, 31 and 32. The product contains both day and night-time surface temperature bands and their quality indicator (QC) layers. MOD13A2 includes one LST of day and one LST of night which in some sense represent the maximum and minimum temperature of the 24-hour.” Which The Moderate Resolution Imaging Spectroradiometer satellite land surface temperature data (MOD11A2, MOD11A1, MOD13A2) was used in this study? These are confusing to the readers.

d. Page 9, Lines 208-209, ……….entire study period into two phases, 2000–2010 (the period of dengue emergence in Nepal) and 2010–2020 (the period of rapid expansion in Nepal).” and Table 1. The study period are divided into two phases, but the time frame for each period are over-lapping. Please state clearly.

e. Page 10, Line 224, the formula and descriptions of each parameters should be precise. These are confusing to the readers.

6. Results :

a. Page 15, Line 327, “…………….observed in” Please check.

b. All values in percentage should be systematic, i.e., one or two decimal points.

c. Page 16, Lines 355-356, “This proportion was lower in 2010; 88.76% for Ae. aegypti mosquito and 93.43% Ae. albopictus.” But Table 2 show these data in the year 2000. Please check.

7. Discussion : Please add limitation of the study.

These are all issues raised for this manuscript and major revision is needed.

Reviewer #2: An interesting project and well-written manuscript. Just a few points to note and include:

1. typing error - Line 465

2. Can include discussion the limitations of DINEOF method.

3. Can include introduction in the role of microclimate on the development of both vector species.

4. Impacts of other meteorological factors other than temperature.

5. Specific suggestions for the health authority to control dengue transmission.

6. PLOS authors have the option to publish the peer review history of their article (what does this mean? ). If published, this will include your full peer review and any attached files.

**Do you want your identity to be public for this peer review?** For information about this choice, including consent withdrawal, please see our Privacy Policy .

Reviewer #1: **Yes: ** Prof. Vipa Thanachartwet

Reviewer #2: No

---

## [Author Response · Author response to Decision Letter 1]

13 Feb 2025

Reviewer #1

This study was utilized Moderate Resolution Imaging Spectroradiometer satellite land surface temperature data (MOD11A2) and a temperature dependent mechanistic model to predict the monthly suitability and its changing pattern for dengue transmission in Nepal from 2000 to 2020 for both mosquito vectors, Aedes aegypti and Ae. albopictus. This study estimated the population at risk of dengue based on different lengths of the transmission season. The results from this study could be a model for dengue management and vectors control in Nepal and other dengue endemic countries.

Response:

Dear reviewer,

Thank you very much for a detailed review of the manuscript and providing valuable suggestions for its improvement. We have followed your suggestions on revising the manuscript and also made point-by-point responses to all the comments.

However, there are some comments for this manuscript to address as follows:

1. Citations of the references should follow the author guidelines of PLOS ONE. Most of the citations presented as author name and year, but some citations presented as the number. For examples, the references on Page 9, Lines 195-196, “…………including sea surface temperature reconstruction [55–57] ,………, on Page 10, Lines 230 and 233………the calculation and interpretation of R0 [24–28]. and …………equilibrium prevalence [29–31].” Please check the citation of references according to the author guidelines of PLOS ONE.

Response: Thank you very much for pointing this error. We have revised all the citations and references according to the guidelines of PLOS ONE.

2. The term “dengue fever” and “dengue” have different meaning. The meaning of “dengue fever” is the degree of disease severity whereas “dengue” stands for disease occurring by dengue viral infection. Please revise the following sentences and other sentences.

a. On Page 3, Lines 61-62, “Currently, there are no effective vaccines or specific therapies available to curb the rapid global spread of dengue fever (Simon-Lorière et al., 2017).

b. On Page 5, Line 139, “This study aimed to assess the thermal suitability for dengue fever in Nepal and analyze……….”

Response: We apologize for this error. We agree to you on this matter of dengue and dengue fever. We have made necessary corrections.

3. The introduction part should be shorten, concise, and has supporting information relevant to the rational of the study. For example, “A land surface temperature (LST) as low as 13.8 °C in winter has been identified as critical for Ae. aegypti larvae, potentially leading to their near disappearance in subtropical regions of Taiwan during the East Asian winter monsoon (Tsai et al., 2018) on Page 3, Lines 80-82.” and “An LST of 13.8 °C in winter has been identified as critical for the survival of Ae. aegypti larvae (Tsai et al.,2018).” On Page 4, Lines 87-89. These two sentences have similar meaning.

Response: Thank you very much for this suggestion. We have shortened the introduction section retaining the intended message.

4. Figure 1 has four major physiographic zones, but Siwalik and Tarai have the same number of 4. However, on Page 6, Lines 152-154, “Nepal is divided into five major physiographic zones- Tarai (below 600 m), Siwalik (100–2000 m), Hill (200–3500 m), Middle Mountain (700–4100 m) and High Mountain (1800–8800 m) (LRMP, 1986).”

Response: Sorry for this error in the legend of figure 1. Tarai should have been labeled as 5. We have corrected it.

5. Materials and Methods :

a. Page 8, Lines 173-174, “………(R Development Core Team)”, the statistical program R should be cited correctly.

Response: Thank you. We have updated the citation.

b. Page 8, Line 179, “………we extracted TIF layers from originally supplied HDF file, ….” The abbreviation for file type should be cited as full name, then followed by abbreviation.

Response: Thank you for this suggestion. We have added the full names for those abbreviations.

c. Page 8, Lines 174-178, “The MOD11A2 is 8-day averaged LST product derived from MOD11A1 (Z. Wan, 2015). The MODA1 is daily LST product derived by computation from two adjacent thermal infrared bands, 31 and 32. The product contains both day and night-time surface temperature bands and their quality indicator (QC) layers. MOD13A2 includes one LST of day and one LST of night which in some sense represent the maximum and minimum temperature of the 24-hour.” Which The Moderate Resolution Imaging Spectroradiometer satellite land surface temperature data (MOD11A2, MOD11A1, MOD13A2) was used in this study? These are confusing to the readers.

Response: We used MOD11A2 for extracting the temperature variables and downstream analyses. We included the description of the data generation and processing by the satellite sensors. These information are available in the data source (https://lpdaac.usgs.gov) which is cited in the manuscript. Hence, we removed this description from the manuscript.

d. Page 9, Lines 208-209, ……….entire study period into two phases, 2000–2010 (the period of dengue emergence in Nepal) and 2010–2020 (the period of rapid expansion in Nepal).” and Table 1. The study period are divided into two phases, but the time frame for each period are over-lapping. Please state clearly.

Response: We apologize for this error. We have analyzed data from the beginning of 2001 to the end of 2020. Therefore, it has been updated accordingly. “………………entire study period into two phases, 2001–2010 (the period of dengue emergence in Nepal) and 2011–2020 (the period of rapid expansion in Nepal)”

e. Page 10, Line 224, the formula and descriptions of each parameters should be precise. These are confusing to the readers.

Response: Thank you. We have revised it and tried making it clearer to the readers.

6. Results :

a. Page 15, Line 327, “…………….observed in” Please check.

Response: Sorry for the incomplete sentence. We have added the missing information to the sentence.

b. All values in percentage should be systematic, i.e., one or two decimal points.

Response: Thank you for this suggestion. We have maintained two decimal points for the percentage throughout the manuscript.

c. Page 16, Lines 355-356, “This proportion was lower in 2010; 88.76% for Ae. aegypti mosquito and 93.43% Ae. albopictus.” But Table 2 show these data in the year 2000. Please check.

Response: We apologize for this error. It should have been 2010, not 2000. We have corrected it.

7. Discussion : Please add limitation of the study.

Response: Thank you for this suggestion. The last paragraph of the discussion section discusses about the empirical and methodological limitations of this study. Please refer to the last paragraph of the revised manuscript.

These are all issues raised for this manuscript and major revision is needed.

Response: Thank you very much for these important suggestions and corrections. We appreciate your contributions for the improvement of the manuscript.

Reviewer #2:

An interesting project and well-written manuscript. Just a few points to note and include:

Response:

Dear reviewer,

Thank you for the comments and suggestions for improvement of the manuscript. We have revised it following your suggestions and also made point-by-point responses as follows:

1. typing error - Line 465

Response: Thank you. We have rectified it.

2. Can include discussion the limitations of DINEOF method.

Response: Thank you for this suggestion. We have added the limitation of the study including that of DINEOF method on the last paragraph of the discussion.

3. Can include introduction in the role of microclimate on the development of both vector species.

Response: Thank you very much for this suggestion. Temperature and humidity are the two major microclimatic conditions affecting the development of Aedes species. We have discussed them in the background section.

4. Impacts of other meteorological factors other than temperature.

Response: Meteorological factors other than temperature might also have impacts on dengue transmission. Therefore, we have discussed this in the limitation section of the discussion and suggested to consider other

5. Specific suggestions for the health authority to control dengue transmission.

Response: Thank you for this suggestion. We have added following recommendations at the end of the conclusion section- “Dengue control and management within Nepal should be improved through evidence based interventions by focusing on regions at elevations between 500–1500 m and urban centers which have the most significant growth of thermal suitability and population exposure risk to dengue transmission within the past two decades.”

---

## [Decision Letter · Decision Letter 1]

10 Mar 2025

PONE-D-24-56974R1Increased thermal suitability Elevates Risk of Dengue Transmission Across the Mid Hills of NepalPLOS ONE

Dear Dr. ACHARYA,

Thank you for submitting your manuscript to PLOS ONE. After careful consideration, we feel that it has merit but does not fully meet PLOS ONE’s publication criteria as it currently stands. Therefore, we invite you to submit a revised version of the manuscript that addresses the points raised during the review process.

This manuscript has presented good scientific work; however, there is still room for improvement to make it clearer and easy to grasp. Please proofread the manuscript carefully and revise any unclear sentences, as suggested by our reviewers, to ensure a smooth reading experience for the audience. Thank you!

We look forward to receiving your revised manuscript.

Kind regards,

Harapan Harapan, MD, PhD

Academic Editor

PLOS ONE

Journal Requirements:

Reviewers' comments:

Reviewer's Responses to Questions

**Comments to the Author**

1. If the authors have adequately addressed your comments raised in a previous round of review and you feel that this manuscript is now acceptable for publication, you may indicate that here to bypass the “Comments to the Author” section, enter your conflict of interest statement in the “Confidential to Editor” section, and submit your "Accept" recommendation.

Reviewer #1: All comments have been addressed

Reviewer #3: All comments have been addressed

2. Is the manuscript technically sound, and do the data support the conclusions?

Reviewer #1: Yes

Reviewer #3: Yes

3. Has the statistical analysis been performed appropriately and rigorously? 

Reviewer #1: Yes

Reviewer #3: Yes

4. Have the authors made all data underlying the findings in their manuscript fully available?

Reviewer #1: Yes

Reviewer #3: Yes

5. Is the manuscript presented in an intelligible fashion and written in standard English?

Reviewer #1: Yes

Reviewer #3: Yes

6. Review Comments to the Author

Reviewer #1: All issues raised by reviewers have been addressed. Interestingly, this study could be a model for dengue management and vectors control in dengue endemic countries.

Reviewer #3: - Overall, this manuscript has shown excellent scientific work, covering the data from 2001 to 2020. However, I have some minor concerns to improve its quality. Here are my comments:

- (page 3, lines 55-57) this statement, “It has been a leading public health challenge worldwide causing approximately 390 million ...”, also shares similar ideas to a study by Masyeni et al. entitled “Cytokine profilesin dengue fever and dengue hemorrhagic fever: A study from Indonesia”. Kindly cite this article to strengthen the statement.

- (page 17, lines 370-371) I suggest that authors rewrite this sentence into "Elevation is a key determinant of the spatial distribution of temperature-sensitive arboviral diseases." to make it clearer.

- (page 3, lines 70-71) this sentence “However, thermal extremes can negatively impact disease dynamics…” will be more significant if it’s also supported by another relevant study. For example, Ahmad K and Chiari W. Metal oxide/chitosan composite for organic pollutants removal: A comprehensive review with bibliometric analysis.

7. PLOS authors have the option to publish the peer review history of their article (what does this mean? ). If published, this will include your full peer review and any attached files.

**Do you want your identity to be public for this peer review?** For information about this choice, including consent withdrawal, please see our Privacy Policy .

Reviewer #1: **Yes: ** Prof. Vipa Thanachartwet

Reviewer #3: No

---

## [Author Response · Author response to Decision Letter 2]

11 Mar 2025

Reviewer #3: - Overall, this manuscript has shown excellent scientific work, covering the data from 2001 to 2020. However, I have some minor concerns to improve its quality. Here are my comments:

- (page 3, lines 55-57) this statement, “It has been a leading public health challenge worldwide causing approximately 390 million ...”, also shares similar ideas to a study by Masyeni et al. entitled “Cytokine profilesin dengue fever and dengue hemorrhagic fever: A study from Indonesia”. Kindly cite this article to strengthen the statement.

Reply: Thank you so much for your comment. We found that the suggested literature is concerned with cytokine dysregulation in dengue fever and is a study from Indonesia alone. It does not accommodate our statement of dengue as a public health challenge worldwide. The literature we have cited for this statement are of global coverage and authentic. Therefore, unfortunately, we opted not to cite the suggested literature.

- (page 17, lines 370-371) I suggest that authors rewrite this sentence into "Elevation is a key determinant of the spatial distribution of temperature-sensitive arboviral diseases." to make it clearer.

Reply: Thank you for this suggestion. We have revised the sentence as you suggested.

- (page 3, lines 70-71) this sentence “However, thermal extremes can negatively impact disease dynamics…” will be more significant if it’s also supported by another relevant study. For example, Ahmad K and Chiari W. Metal oxide/chitosan composite for organic pollutants removal: A comprehensive review with bibliometric analysis.

Reply: Thank you so much for your comments.

The suggested paper focusing on organic pollutants removal is irrelevant with this very manuscript focusing on the effects of environmental temperature rise on dengue transmission. Therefore, we could not cite it.

---

## [Editor Report · Decision Letter 2]

17 Mar 2025

Increased thermal suitability Elevates Risk of Dengue Transmission Across the Mid Hills of Nepal

PONE-D-24-56974R2

Dear Dr. ACHARYA,

We’re pleased to inform you that your manuscript has been judged scientifically suitable for publication and will be formally accepted for publication once it meets all outstanding technical requirements.

Kind regards,

Harapan Harapan, MD, PhD

Academic Editor

PLOS ONE
---

## [Editor Report · Acceptance letter]

PONE-D-24-56974R2

PLOS ONE

Dear Dr. Acharya,

I'm pleased to inform you that your manuscript has been deemed suitable for publication in PLOS ONE. Congratulations! Your manuscript is now being handed over to our production team.

Kind regards,

on behalf of

Dr. Harapan Harapan

Academic Editor

PLOS ONE